# Data Scaling Laws in NMT: The Effect of Noise and Architecture

## Abstract

In this work, we empirically study the data scaling properties of neural machine translation (NMT). We first establish that the test loss of encoder-decoder transformer models scales as a power law in the number of training samples, with a dependence on the model size. We then systematically vary various aspects of the training setup to understand how they impact the data scaling laws. In particular, we change the (1) Architecture and task setup, to a Transformer-LSTM Hybrid as well as a Decoder-only transformer with language modeling loss (2) Noise level in the training distribution, starting with noisy data with filtering applied as well as clean data corrupted with synthetic iid noise. In all the above cases, we find that the data scaling exponents are minimally impacted, suggesting that marginally worse architectures or training data quality can be compensated for by adding *more data*. Lastly, we find that changing the training distribution to use back-translated data instead of parallel data, can impact the scaling exponent.

## 1    Introduction

In deep learning, a reliable recipe to improve the generalization performance of a model is to increase the amount of compute and data used to train it (Krizhevsky et al., 2012; Brown et al., 2020). Many recent advances in deep learning are directed towards increasing the efficiency of digesting data through advancements in architecture (Vaswani et al., 2017), improving the quality of the data with filtering, or incorporating entirely new sources of data into training (Radford et al., 2021; Chen et al., 2020). In this new paradigm of machine learning, it is crucial to understand the data-efficiency of our training methods, and whether improvements to them at small scale translate into improvements at large scale.

Recent work (Hestness et al., 2017; Rosenfeld et al., 2019; Kaplan et al., 2020) on *scaling laws* offers a useful tool to answer such questions — they show that the test loss of a model scales predictably and smoothly as a power law in the relevant quantities of interest such as dataset size ($D$), model size ($N$) and amount of compute ($C$). We take these findings a step further and ask how different interventions to the training setup impact the data scaling curves in the setup of neural machine translation (NMT):

*How do changes to the training setup (such as architecture, noise) impact the data scaling curves in NMT?*

Practically, data scaling laws can be leveraged to make experimental decisions for future large scale experiments, and to decide where computational and research efforts should be focused. Currently, NMT models are trained using massive web-scale data; Arivazhagan et al. (2019) used 25B training examples (approx.1T tokens) and with the advent of the self-supervised learning techniques (Liu et al., 2020; Raffel et al., 2020; Siddhant et al., 2020) this number can easily reach 10+T tokens. At such large scales of data, it is unfeasible to 'just perform the experiment', and scaling laws can be used to drive training decisions. For instance, if small changes in architecture do not lead to a change in scaling exponent (as shown in Figure 1B), then architecture choice can be driven by other factors such as computational efficiency by paying a small penalty in the form of more data. In the opposite scenario, if a training setup performs worse at a given dataset size but scales much better, it should be chosen for future larger scale experiments.

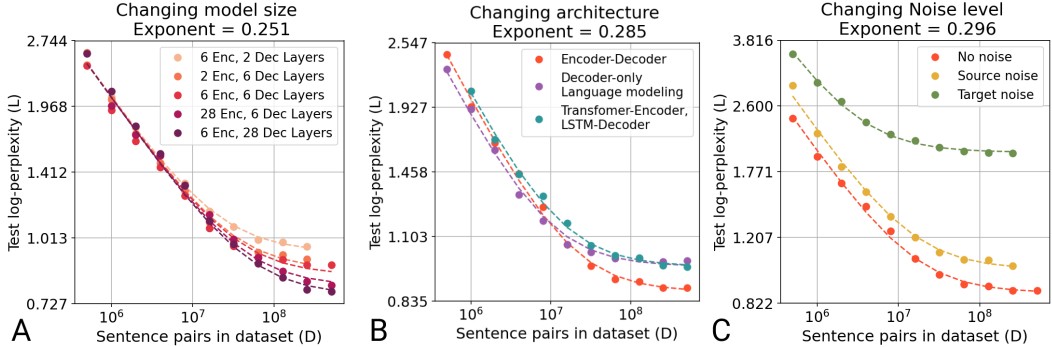

Figure 1: **Data scaling exponent is minimally impacted by changes to the training set up** We train a series of models on increasing amounts of data $\{500K...512M\}$ sentence pairs while (A) Changing the depth of the encoder-decoder transformer (B) Changing the architecture and task set-up (C) Changing the noise in the training distribution. For each figure, we fit the data scaling law similar Eq. 1 and find that a common exponent $p$ provides good fits for the empirical observations. Full details in Section 2, 3 & 4 respectively.

## 1.1 OUR CONTRIBUTIONS

In this work, we first establish that the test log-perplexity of encoder-decoder transformer models trained to perform neural machine translation (NMT) follows a power law in the dataset size, with a dependence on the model size (Figure 1A). We demonstrate that our scaling law predicts experimental results over 3 orders of magnitude (from 500K-512M) training examples (sentence pairs in NMT) [1]. Then, we systematically vary aspects of the training setup to understand how these impact the scaling laws.

In particular, we consider the effect of (1) Changing architecture and task setup from a vanilla Transformer Encoder-Decoder to Transformer Enoder-LSTM Decoder Hybrid as well as a Decoder-only Transformer with language modeling objective (2) Changing the level of filtering in the dataset (3) Changing the amount of noise on the source (input) side and the target (output) side (4) Using back-translated data. Surprisingly, we find that, with the exception of back-translation, these changes *do not impact the scaling exponents much* (See Fig. 1 for example of changing model size, architecture and level of noise). Some of these changes affect the *bias* or the final loss value at infinite data.

Our work suggests that many of the common operations used to boost performance, such as small changes to the architecture or data filtering, can be replaced by adding an additional constant factor of data. That is, **in some cases, sub-optimalities in the architectures and noise in the datasets can both be considered as (model and data) scaling penalties**.

## 1.2 EXPERIMENTAL SETUP

**Models:** Our experiments are conducted on pre-layer transformer networks (Xiong et al., 2020). Models are trained with per-token cross-entropy loss and Adafactor optimizer (Shazeer & Stern, 2018). All models are trained with a fixed batch-size of 500K tokens and dropout rate of 0.1 for residuals, feed-forward activations and attention. For the small dataset sizes, the models are trained to early stopping (as measured on the log-perplexity of a held-out development set) and for large dataset sizes they are trained for up to 500K gradient steps. The hyperparameters for these models were optimized for a 6 encoder layer and 6 decoder layer model trained on 2.2 billion sentence pairs. We train 5 different model sizes {2L6L, 6L2L, 6L6L, 6L28L, 28L6L}, where 28L6L means 28 encoder layers and 6 decoder layers.

We also train two decoder-only models with a language modeling loss with {9L, 13L}, and three Hybrid-LSTM model with {6L2L, 6L6L, 6L12L}. All the hyperparameters are matched as closely as possible between these models to provide an apples-to-apples comparison.

---

[1]Corresponding to 27.6 billion tokens.

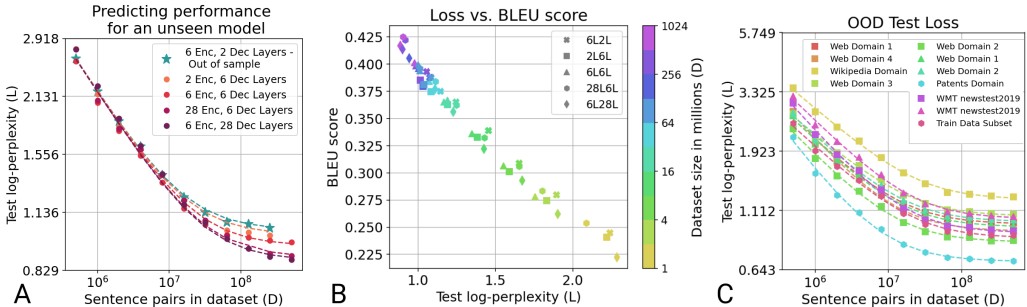

Figure 2: (A) Using the joint data and model scaling law from Eq. 2 to predict the performance of a previously unseen model. (B) BLEU score and test loss display an almost-linear relationship (C) Scaling law for different OOD test sets for a 6L6L model (Results plotted for Web Domain 1 test set.)

**Training data** In our experiments, the models are trained on two large scale datasets. The first set of experiments on data size scaling are using the dataset sampled from an in-house parallel corpora containing up to 2.2B sentences translated from English to German. We sample training datasets of sizes {1M, 2M, 4M, 8M, 16M, 32M, 64M, 128M, 256M, 512M} independently to study the data scaling laws. The second set of experiments are conducted with Paracrawl dataset (Bañón et al., 2020) and with and without filtering applied. The details are described in Section 4.

**Test data** The model performance is measured on a held-out dataset from the training distribution. Additionally, we also measure performance on various out-of-distribution test datasets. These test datasets have different domain composition and sampling methods. The domains considered are (i) Web-Domain (ii) News-Domain (iii) Wikipedia and (iv) Patents. The news-domain test sets come from the WMT2019 evaluation campaign (newstest2019) for all language pairs. The other test sets are internal test sets representing the different domains, ranging from 500 to 5000 sentence pairs for each set.

## 2 DATA SCALING LAWS

We begin our investigation by training a series of large scale encoder-decoder transformer models on the in-house English to German parallel corpus, with the parameters ranging from 170M to 800M. Our dataset sizes range from 500K to 512M sentence pairs, covers 28B tokens, which is 10 times larger than the experiments conducted in prior work on scaling laws for NMT Gordon et al. (2021). Our goal is to find a function in the relevant variables that predicts test performance of our experiments. We will mainly focus on the test log-perplexity on a heldout dataset from the training distribution, but we will also discuss scaling of BLEU scores (Section 2.1) and performance on other out-of-distribution test sets (Section 2.2) in later sections.

**Form of the scaling law:** The chosen scaling law must exhibit decreasing loss with dataset size $D$, and at infinite data $D = \infty$, the model must converge to a finite constant. Additionally, Kaplan et al. (2020); Bahri et al. (2021), conjecture that when $D \to \infty$, the models are in a "variance-limited" regime and the loss should scale as $O(1/D)$. These desiderata are satisfied by the following scaling law for a fixed model size:

$$L = \alpha \left( \frac{D_0}{D} + C \right)^p \tag{1}$$

where $\alpha$ is a multiplicative constant, $p$ is the scaling exponent and $C$ is a model-dependent constant that are fitted empirically ($D_0 = 1e6$ is a fixed normalization constant use for numerical stability). We find that this scaling law indeed provides a good fit for the experimental observations. Moreover, when we fit just the last few datapoints (See Figure 5), we find that the loss scales as $O(1/D)$, thus confirming the conjecture even for large-scale supervised learning setup.

Now as we change the model size (depth, width, or depth-width aspect ratio etc), it is not apriori obvious if these different models, say encoder-heavy vs. decoder-heavy models, should have the same data scaling parameters $\alpha, p$. For instance, Ghorbani et al. (2021) observe that parameter scaling for encoder-decoder models depends separately on the number of encoder and number of decoder models. Thus, we train 5 different models {2L6L, 6L2L, 6L6L, 28L6L, 6L28L} including very asymmetric ones and find that their scaling parameters $\alpha, p$ are in fact very similar (See Appendix A.1 for individual fits for Equation 1 to each model). Additionally, the constant $C$ must depend on the model size, since at infinite data $D = \infty$ different models will converge to different loss values $\alpha C^p$, with larger models converging to lower loss values. Figure 1A shows the fit for a common $\alpha, p$ and 5 different $\{C_i\}_{i=1}^5$ for each different model. We find that a common exponent is sufficient to provide a good fit for the experimental data.

We can go a step further and replace the constant $C$ such that it recovers the parameter scaling curve at $D = \infty$. To do so, we leverage the parameter scaling law found by Ghorbani et al. (2021) and fit the following final scaling law:

$$
L(D, N_e, N_d) = \alpha \left( \frac{D_0}{D} + \beta \Big( \frac{1}{N_e^{p_e}} \frac{1}{N_d^{p_d}} + L_\infty \Big)^{1/p} \right)^p
\tag{2}
$$

The only free parameters in this equation are $\alpha$ and $p$, and the parameters in the right term $(\beta, N_e, N_d, L_\infty)$ are directly borrowed from Ghorbani et al. (2021), such that they converge to the parameter scaling law at $D = \infty$. The fit using this scaling law is shown in Figure 2A. Since we have a joint scaling law with dataset size and parameters, we can use this to predict the test loss of models that were not used to fit the scaling law. For example, in Figure 2A we find $\alpha, p$ for all models while holding out the model with 6L28L and later we are able to predict the performance of this *out-of-sample* model. Note that this scaling law differs from Gordon et al. (2021) in two ways (1) Using the encoder and decoder parameters separately vs. using the total number of parameters (2) An additional $L_\infty$ term[2] that remains even when the $N_e, N_d, D = \infty$.

**Implications:** Equation 1 suggests that there exists two operating regimes for data scaling: (i) data-limited regime where $\frac{D_0}{D} \gg C$, and (ii) capacity limited regime where $\frac{D_0}{D} \ll C$. Fitted exponents in Figure 1 suggest that, in the data limited regime, loss scales as $O(D^{-1/4})$, suggesting a marginal value of $O(D^{-5/4})$ for additional data. Increasing the model capacity in this regime has negligible impact on the loss. In the capacity limited regime however, the loss scales as $O(D^{-1})$ suggesting a (significantly smaller) marginal value of $O(D^{-2})$ for additional data. In this regime, the loss value is dominated by the model-dependent constant $C$ and most of the improvement can be had by increasing the model size. There is a smooth *phase transition* between these two regimes at approximately $\frac{CD}{D_0} = 1$ (See Appendix C for an illustration). Thus, by increasing the model size (which reduces $C$), one can push the transition to larger values of $D$ and leverage the available data more efficiently.

## 2.1 BLEU SCORE

Language tasks can roughly be categorized into two groups *understanding* tasks where a given piece of text is tasked to be encoded for downstream classification (eg. sentiment analysis, named entity recognition), and *generation* tasks where a representation of a piece of text is used (conditioned) to generate another arbitrary length sequence of text (eg. summarization, question answering). Machine translation, without loss of generality, belongs to both of the categories: source content first needs to be understood and the corresponding target sequence must be generated. Given this categorization, we not only care about the model score on reference target sequence (measured in log-perplexity) but we also care about the generation quality. Once sequences are generated from a sequence model, in autoregressive fashion or not, we resort to automatic measures like BLEU

---

[2]Gordon et al. (2021) use relatively smaller model sizes. At small $N_e, N_d$, the $L_\infty$ term is dominated by the $1/N_e^{p_e} N_d^{p_d}$ term, which may explain why they did not need to use an additional $L_\infty$ term

score.[3] We find that the BLEU score[4] and test log-perplexity have a nearly linear relationship as shown in Figure 2B. Note that this linear relationship is known to break down at very low loss values (Ghorbani et al., 2021).

## 2.2 OOD TEST DATASETS

Practically, we are also interested in the performance of our models on out-of-distribution (OOD) test sets, outside the heldout set. These test sets are related to the training distribution but may have different support or composition. To understand how performance on such test sets scales with dataset size, we measure the performance on the test sets as described in Section 1.2. We find that the test loss on these test sets also follows a similar power law in the dataset size as Equation 1. Figure 2C shows the test loss fits for all the test sets for a 6L6L model. We find that most of the test sets have similar scaling exponents (See Figure 6C for exact values). That is, we do not observe any major differences in scaling on the basis on the test set composition — both target-original and source-original test sets scale similarly. Additionally, since the different test sets have similar scaling exponents, this implies that the in-distribution loss and the out-of-distribution loss must have a nearly-linear relationship (See Figure 6C). This is in line with previous findings in vision (Miller et al., 2021). Why these distributions scale similarly (or why they have a linear relationship) is still an open theoretical question.

## 3 THE EFFECT OF ARCHITECTURE

Innovations in architecture have been fundamental to progress in deep learning. It is often suggested that architectures confer certain inductive biases that enable learning. However, recent trend in machine learning largely confirms the trade-off between architectural biases and the amounts of data, namely tendency towards architectures with low inductive biases are being trained on large amounts of data, compute and parameters. Thus, we believe it is essential to determine whether these "low bias" architectures have similar scaling behavior with respect to the dataset size.

To understand the relationship between the amounts of data and architectural bias, we pick three common architecture and loss set ups that are commonly used for machine translation. We take an encoder-decoder transformer described in detail in Section 2 as the baseline. Next, we pick a Hybrid architecture with a transformer encoder and an LSTM decoder (Chen et al., 2018) due to their wide adoption by the industry applications.[5] This allows us to compare the scaling of a transformer vs. an LSTM — both popular but different sequence-to-sequence architectures. Finally, we use a decoder-only transformer that is trained with a language modeling (LM) loss. Thus, the last model changes not only the architecture, but also the loss (by including an LM loss on the source side). This setup is identical to GPT-3, and thus allows to compare if these two different set ups for doing translation have similar data efficiency.

To compare the scaling of these models, we train a series of models on increasing subsets of the data. All the three models have $\sim 300M$ parameters. Then, we fit a scaling law with a common exponent $p$, but model-dependent $\{\alpha_i, C_i\}_{i=1}^3$. The results of our experiment are shown in Figure 1B. The fitting parameters are shown in Appendix D. We find that this scaling law with a common exponent

---

[3]While we report BLEU in addition to log-perplexity scores in our study, we acknowledge and would like to bring the recent findings on the deficiency of BLEU as an automatic metric to the readers attention. As the MT systems got better over the years, BLEU scores (along with several other automatic metrics) started to lose their sensitivity to approximate human judgement (Zhang & Toral, 2019; Mathur et al., 2020; Freitag et al., 2020; 2021; Kocmi et al., 2021) and the translation community has started to experiment with learned metrics such as COMET (Rei et al., 2020) or BLEURT (Sellam et al., 2020). We find the ongoing discussions around the established automatic metrics quite important, but refrain from dilating the scope our study, hence report both model scores and BLEU scores.

[4]BLEU score is a precision based metric that compares a reference translation with the generated hypothesis by the model and yields a score between 0 and 1, taking into account n-gram overlap between reference and hypothesis while compensating for the lack of recall with a brevity penalty.

[5]See https://www.microsoft.com/en-us/translator/blog/2019/06/17/ neural-machine-translation-enabling-human-parity-innovations-in-the-cloud and https://ai.googleblog.com/2020/06/recent-advances-in-google-translate. html

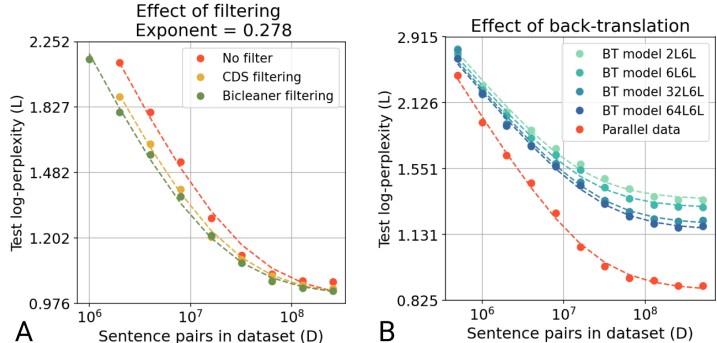

Figure 3: (A) **Effect of filtering:** We apply two different filtering algorithms to the raw Paracrawl dataset and evaluate the data scaling curve. We find that a common exponent provides a good fit for the experimental observations. (B) **Effect of back-translation:** We train a 6L6L model on back-translated data from 4 different back-translation models {2L6L, 6L6L, 32L6L, 64L6L}. We find that the scaling exponent for back-translated data is worse than that for clean parallel data.

describes the observed experimental data well. Please refer to Appendix D for more results with different model sizes of the Hybrids and Decoder-only models.

Our experiments show that while subtly different architectures may have different performances on a fixed dataset size, the architectures may nevertheless scale similarly. Thus, *we can compensate for a marginally worse architecture by adding more data*. Crucially, the factor of additional data to be added does not depend on the loss value, it will always be $\frac{\alpha_1}{\alpha_2}^{1/p}$, where $\alpha_1, \alpha_2$ are the multiplicative constants for the two architectures and $p$ is the common scaling exponent. On the other hand, if the exponents were different, the factor of data to be added to achieve equal performance would *increase* exponentially with decreasing loss. This suggests that when choosing between multiple architectures that have similar data scaling, the decision can freely be driven by other considerations such as compute efficiency, multi-task abilities, compressibility for deployment or generation latency etc.

## 4 THE EFFECT OF NOISE

Large scale parallel corpora are essential to building high-quality machine translation models (Arivazhagan et al., 2019; Lepikhin et al., 2020; Fan et al., 2020). Such datasets are created by crawling web pages and performing various post-processing steps to create parallel data such as document alignment and sentence alignment (El-Kishky et al., 2020; Bañón et al., 2020). However, there are many ways in which noise can enter this pipeline — misaligned sentences, copied URLs, typos, mistranslations and so on. Such noise can potentially be detrimental to translation quality. For instance, prior work (Khayrallah & Koehn, 2018) has found that adding a large amount of noisy data to high quality parallel data can have a catastrophic effect on the performance of NMT models. In this section, we will study the effect of noise, not just by comparing noisy and clean data at a single dataset size, but by comparing their scaling for increasing dataset sizes. We approach this question in two opposite and complementary ways (1) We start with a noisy web-crawled corpus, Paracrawl English to German (Bañón et al., 2020) and apply filtering algorithms to it (Section 4.1) (2) We start with a clean parallel corpus and add different types of noise to it (Sections 4.2.1, 4.2.2).

### 4.1 DATA FILTERING

Due to the prevalence of noise in web-crawled corpora and its impact on machine translation models, a variety of algorithms and heuristics have been developed to filter out noisy sentences (Wang et al., 2018; Junczys-Dowmunt, 2018; Ramírez-Sánchez et al., 2020). Our goal here is to understand the impact of data filtering on the data scaling properties of NMT.

**Our setup:** To understand the effect of data filtering, we use the largest publicly available Paracrawl English-German dataset(Bañón et al., 2020). We lightly filter the raw dataset with de-duplication, length filtering ($\leq 256$) and language ID filtering. We also remove near duplicates of our test sets

with a 10-gram overlap. This leaves $\sim 750M$ noisy sentence pairs. We train a 6L6L model on this dataset with increasing dataset sizes $\{1M, 2M...256M\}$. To study the effects of the filtering, we use two filtering algorithms. For the first filtering algorithm, we use the Bicleaner scores (Ramírez-Sánchez et al., 2020) that are publicly released along with the Paracrawl dataset v8.1. This is a score ranging from $(0, 1)$, with a higher score indicating higher sentence quality. The bicleaner scores includes various hard-coded rules, language-model fluency scores, and scores from a classifier trained to detect mutual translations. We use a threshold of $0.5$ and discard all sentences below this threshold leaving us with $\sim 300M$ sentence pairs. For the second filtering algorithm, we choose Contrastive Data Selection (CDS) (Wang et al., 2018), which belongs to a family of cross-entropy-based filtering algorithms (Moore & Lewis, 2010; Junczys-Dowmunt, 2018). CDS scores the quality of each sentence pair according to the difference in cross entropy scores between two related translation models: a clean model that was fine-tuned on a trusted dataset, and a noisy one that was not. We choose the top 50% of the CDS-ranked sentence pairs.

**Results:** The results for the data scaling law for all three training datasets are shown in Figure 3A and fitted coefficients in Table 1. We measure the test log perplexity on the Web Domain 1 test set. We fit a scaling law with a common exponent $p$ and separate $\alpha, C$ for each training set. We find that a common exponent fits the experimental data well. We make the following observations:

*Multiplicative constant $\alpha$ shift:* The filtered dataset has lower $\alpha$, implying that at a given dataset size, the loss for filtered data is lower. Thus, if you were constrained by compute to use a small fixed dataset size, it would be advisable to use a filtered dataset.

*Loss at convergence $\alpha C^p$ is the same:* In our experiments, we find that the three datasets converge to the same loss for a 6L6L model. This implies that at large dataset sizes, a 6L6L model is unable to distinguish the differences be-

|  | $\alpha$ | $C$ | $p$ |
|---|---|---|---|
| No filter | 2.501 | 0.034 | |
| CDS | 2.235 | 0.054 | 0.278 |
| Bicleaner | 2.130 | 0.064 | |

Table 1: Coefficients for filtering

tween a filtered and unfiltered dataset. We would like to emphasize that this is a consequence of the particular filtering algorithms used. For instance, it is possible to construct a filtering algorithm that is very biased and discards all data from a certain domain, which may converge to a higher loss value.

*Similar exponents $p$:* We find that a common exponent is sufficient to describe the experimental data. This finding has the following important implication: We can compensate for noise in the dataset simply by adding more data!

There is no standard, task-independent definition or measure of sentence quality or noise. Thus, any filtering algorithm always runs the risk of biasing the training dataset, say, towards the in-domain trusted dataset used for filtering. For example, a recent study by Gao (2021) shows that very aggressive filtering can negatively impact the downstream performance of language models. Our results show that the effect of unfiltered noisy sentence pairs is *not* catastrophic — while ***some amount of filtering may be desirable for computational efficiency, we can replace filtered data with more unfiltered data***.

## 4.2 ADDING NOISE

While understanding the effect of filtering on data scaling curves is practically informative, filtering combines many different types of noise and heuristics together. To get a more detailed understanding of the effect of noise, we now *add* noise to a clean dataset to have finer control over the dataset. In particular, we make two different types of distinctions. First, we add noise either only to the source side (the input sentences) or only to the target side (the output sentences). Second, we consider *independent vs. dependent noise*. Independent noise, includes noise such as changing a character to a random character or deleting a random word — the noise does not depend on the source/target sentence itself. By dependent noise, we mean that noise added to the sentence depends on the sentence itself, for eg: if the word 'cat' is always mistranslated. We believe that this is a natural distinction since the effect of the former type of noise can (at least information-theoretically) be reversed if we are provided with more data. On the other hand, dependent noise can bias the distribution in more drastic, irreversible ways.

### 4.2.1 Independent noise

We start with a simple setup of the following types of iid noise added to the source and target side respectively: (1) *Character level:* We perturb $p = 0.1$ fraction of the characters in the sentences to random characters (alphumeric + punctuations), (2) *Word level:* We delete $p = 0.15$ fraction of the words from either the source or the target side, and (3) *Sentence level:* For $p = 0.1$ fraction of the sentences, we shuffle the mapping sentences of the sentence pairs. These noise types

|  | $\alpha$ | $C$ | $p$ |
|---|---|---|---|
| No noise | 1.969 | 0.064 | |
| Source noise | 2.222 | 0.067 | 0.296 |
| Target noise | 2.772 | 0.323 | |

Table 2: Coefficients for added noise

have also been considered in prior work Khayrallah & Koehn (2018). Next, we train a 6L6L transformer model on increasing subsets of the noisy training datasets. The results are shown in Figure 1C and Table 3.

Our first observation is that, similar to the case of data filtering, we can fit a power law of Equation 1 with a common exponent $p$, but different $\alpha, C$ for the different training sets to our experimental observations. On the other hand, unlike filtering, both the source and target noise datasets do not converge to the same loss value as the clean dataset at large dataset sizes $D \to \infty$. These results show that while the exponents for different datasets can be similar, it is also important to consider the $C$ or the loss where these models converge. In this particular case, more data cannot always offset the effect of noise. It is an open question if this is because the model size 6L6L is too weak to learn in the presence of this amount of noise, that these types of noise are disruptive to neural network training irrespective of model size, or that this problem is hard to solve at finite samples for any class of models. It is also an interesting open question to see if we can predict which types of noise will lead to a different $C$.

Lastly, another important observation is that target noise is much more harmful than source noise for the same amount of noise. This may help explain why backtranslation (Sennrich et al., 2016), a widely used data augmentation technique for machine translation, works — since it it uses a clean monolingual target corpus and noisy back-translated source sentences. This is in contrast with observations in the vision domain Bahri et al. (2021), where changes to the input distribution change the exponent, but changes to the output distribution keep the exponent unchanged.

### 4.2.2 Dependent noise: Back-translation

Now we turn our attention to changing the training distribution by using back-translated data instead of parallel data. Back-translation (BT) (Sennrich et al., 2016) is a common data-augmentation technique employed in MT to increase the amount of training data. Say you are training your model to translate English to German sentences. With back-translation, one would use a reverse model trained from German to English, and a clean monolingual German corpus to generate English-German sentence pairs. Back-translation can be considered a type of noise that is added to the source side. However, it differs from the independent noise considered in Section 4 in that noise depends on the source/target sentence itself. For instance, if the BT model was never trained on any sentences on the topic of animals, it will make systematic errors on such sentences. These errors can be impossible for the downstream model to reverse, no matter how much training data is provided to it. This makes it an interesting setup to study data scaling, as it is not apriori obvious if such a distributional change would impact just the bias of the model $C$ or also the scaling exponent $p$.

**Setup:** In our setup, to minimize the number of confounders, we extract the same German target side of the same dataset that was used to train our models in Section 2. This keeps at least the target distribution the same as that in Sections 2 and 4. We use four different German to English encoder-decoder models of sizes {2L6L, 6L6L, 32L6L, 64L6L} to translate the German sentences, thus giving us four different datasets with English source sentences of varying quality, with the smallest model producing the 'noisiest' source sentences. Note that these German to English models are trained on a different dataset, although there may be some sentence overlaps with the English to German datasets. To evaluate the data scaling, we train a 6L6L model on increasing dataset sizes for all these datasets.

The results of our experiments are shown in Figure 3B. We fit a scaling law with common $\alpha, p$ and dataset dependent $C$. Interestingly, we find that the scaling exponent of

the BT trained models is slightly lower ($\sim$ 0.2) than the scaling exponents of the parallel dataset ($\sim$ 0.28) as is also visible in Figure 3B. We did not get good fits when tried to jointly fit the BT data with the clean parallel data with a common exponent. Moreover, the BT datasets converge to a worse loss value at the infinite data regime.

Another point of note is that different BT model sizes have similar scaling, but larger models converge to marginally better test log-perplexities. Thus, if we are in the extreme data-limited regime, almost any back-translation model will provide good improvements, but as we approach very large dataset sizes, it will be more beneficial to use a larger BT model.

|  | $\alpha$ | $C$ | $p$ |
|---|---|---|---|
| BT model 2L6L | 2.343 | 0.059 | |
| BT model 6L6L | 2.288 | 0.054 | 0.0198 |
| BT model 32L6L | 2.251 | 0.040 | |
| BT model 64L6L | 2.224 | 0.037 | |
| Parallel data | 1.196 | 0.048 | 0.271 |

Table 3: Coefficients for BT

Taken altogether, these three experiments show that noise has less impact than one might have expected on the exponent characterizing the data-scaling behavior of NMT, given the strong emphasis on data quality in the NMT community. Both filtering natural noise and adding artificial *independent* noise have no impact on the exponent. However, they do impact the multiplicative constant, meaning that for a fixed computation or data budget, filtering remains quite relevant. Crucially, the techniques outlined here give practitioners tools they can use to help determine when effort should be put into improving removing noise, and when they should focus on collecting more data.

## 5 RELATED WORKS

Our work builds extensively on prior work on scaling laws Hestness et al. (2017); Rosenfeld et al. (2019) and in particular Kaplan et al. (2020). The papers most closely related to our work are Hestness et al. (2017); Gordon et al. (2021) who provide data and parameter scaling laws for NMT. In addition to conducting experiments at a much larger scale, our work goes beyond the experimental setup in these papers and considers the question of how various change to the training setup impact the data scaling law. Additionally, the form of our scaling law departs slightly from the scaling laws presented in Gordon et al. (2021) as discussed in Section 2. Our scaling law differs from Hestness et al. (2017) in that they conduct experiments with LSTMs and their law does not scale as $O(1/D)$ when $D \to \infty$. Despite these differences, the scaling exponents found in both these papers are in the same range as ours $\sim 0.3$. Gordon et al. (2021) also conduct experiments on multiple language pairs and find that their exponents are in the same range. Note that these exponents are much higher than those found for the unconditional language modeling case by Kaplan et al. (2020); Hestness et al. (2017) (0.1 vs. 0.3). Hoiem et al. (2021) conduct an analysis of data scaling curves in the vision domain with a focus on architectures and pre-training.

## 6 DISCUSSION AND CONCLUSIONS

In this work, we study the data scaling of models trained to perform neural machine translation under various interventions to the training pipeline, including changes to architecture as well as the training distribution. We find that a majority of these changes lead only to a multiplicative shift in the scaling curves, and the exponent changes minimally. Practically, this suggests that many advancements that seem significant at smaller scale, could be equivalently achieved by moving further across the data scaling curve and adding more data.

Apart from the practical implications, this work also raises some interesting theoretical questions. If so many interventions to the training pipeline keep the exponents unchanged, then this may be indicative of a deeper commonality in the mechanism by which these deep networks learn. For instance, recent work (Bahri et al., 2021; Sharma & Kaplan, 2020) conjecture that the data scaling exponent captures the "dimension of the data manifold" as it is represented by the model. If this is the case, then our experiments suggest that some changes to the architecture or certain types of changes to the distribution do not change this 'manifold'.

In conclusion, we believe that understanding how different training methods change the scaling of the test performance can yield important insights both theoretically and practically.

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

## A  DETAILS ABOUT POWER LAW FITTING

### A.1  SEPARATE FITS AND VARIATION IN THE EXPONENT

In Section 2, we showed that a single common exponent $p$ gives a reasonable fits to the experimental observations for different model sizes. We now examine the difference in the co-efficients for each of these models. To do so, we fit a separate power law from Equation 1 to each model as shown in Figure 4A. The scaling exponents for all models are not exactly the same — they do have minor differences.

Some of these variations can be attributed to the sensitivity of these scaling parameters to randomness in the training procedure. There are multiple sources of randomness in training — initial random seed, randomness over sampling of the training set, randomness in SGD training such as batch order. To understand the effect of these, we sample 5 different versions of the training set for dataset sizes $\{250K, 500K, 1M\}$ (we choose these as they require low compute to train to convergence) and train networks on these datasets from scratch. Figure 4B shows the standard deviation observed in the test loss for the 6L6L and 6L28L models. As we can see, the variance is up to 2% of the loss. The variance in larger datasets is expected to be lower than those for smaller datasets.

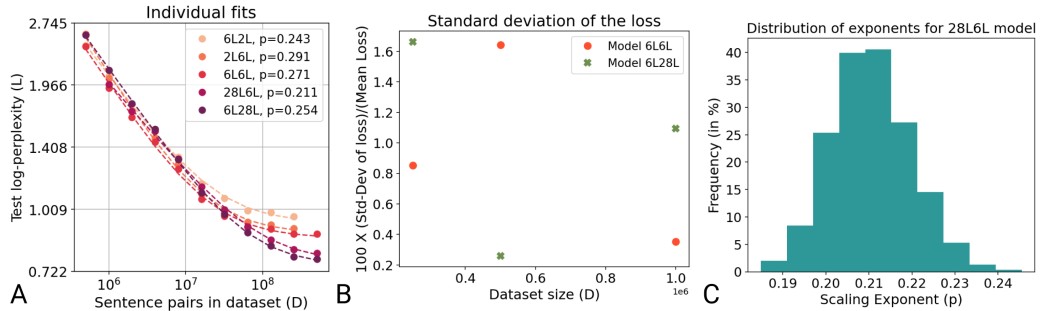

Figure 4: (A) Separate power law fits from Equation 1 to different model sizes. We observe small differences in the scaling co-efficients (B) Standard deviation in the loss due to randomness in training. The loss varies by up to 2% (C) Distribution of the scaling exponents from a monte-carlo simulation assuming a 2% standard deviation in the loss.

To understand how this variance would affect the final observed scaling co-efficients, we do the following monte carlo experiment: We assume that the loss is distributed as $\mathcal{N}(l, 0.02l)$ where $l$ is the loss for a given dataset size and model. We then simulate different loss values from this distribution for all the dataset sizes, and fit scaling co-efficients to it. This gives us a distribution over the scaling co-efficients. Figure 4C shows the distribution of the scaling exponents obtained from this procedure for the 28L6L model. As we can see, a 2% randomness in the test loss, gives us a standard deviation of 0.02 in the scaling exponent. This provides a benchmark in comparing exponents obtained from two different experiments (say two different architectures). Please note that this is only a rough calculation that is meant to give some intuition for how these sacling parameters may be affected by various sources of randomness.

### A.2  VARIANCE-LIMITED REGIME

We fit the scaling law $L = \gamma(D_0/D)^p + B$ to dataset sizes $>= 32M$. If the "variance-limited" conjecture is correct, then the scaling exponent should be $p \sim 1$. Figure 5 indeed shows that this is the case. Moreover, the transition point to this regime should occur later for larger model sizes. As we can see, the exponents for the larger models has not reached 1.

## B  SCALING LAWS FOR DIFFERENT DATASETS

In all our main experiments, we study the scaling laws for a heldout test set from the same distribution as the training set. But, practically, we may be interested in the performance of some other out-of-distribution test sets. These other test sets have some overlap with the training distribution but

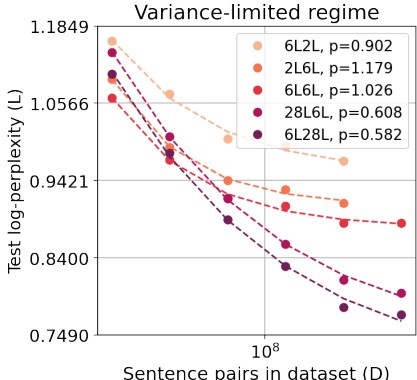

Figure 5: Power law fits for large dataset sizes show that the loss decays as $O(1/D)$

may have different support or composition. To understand how performance on such test sets scales with dataset size, we measure the performance on various other test sets as described in Section **??**.

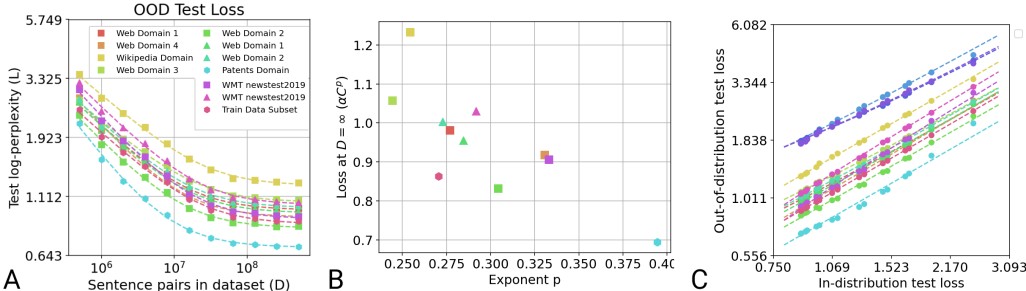

Figure 6: Scaling laws for various OOD test sets for 6L6L model

We find that the test loss on these additional test sets also follows a similar power law in the dataset size as Equation 1. Figure 6 shows the test loss fits for 14 different test sets for a 6L6L model. We find that most of the test sets have similar scaling exponents. That is, we do not observe any major differences on the basis on the test set composition — both target and source original test sets scale similarly.

Additionally, since the different test sets have similar scaling properties, this implies that the in-distribution loss and the out-of-distribution loss must have a nearly-linear relationship.

## C  DATA SCALING PHASE TRANSITION

We fit the scaling law shown in Equation 1.

$$L = \alpha \left( \frac{D_0}{D} + C \right)^p \tag{3}$$

This equation displays two scaling regimes:

1. Over-parameterized (or small $D$): In this regime, the $D_0/D$ term dominates the loss and the loss scales as $O(1/D^p)$.

2. Under-parameterized (or large $D$): In this regime, we can take a Taylor's approximation for the small term $D_0/D$ which leads to the loss scaling as $O(1/D)$.

Equating the derivatives of the two expressions provides an expression for the point where the marginal value of data transitions (and hence the model moves to capacity limited regime). A simple calculation shows that this point occurs at $CD = D_0$. Figure 7 provides an illustration for this transition in the simple scenario where $D_0 = 1$.

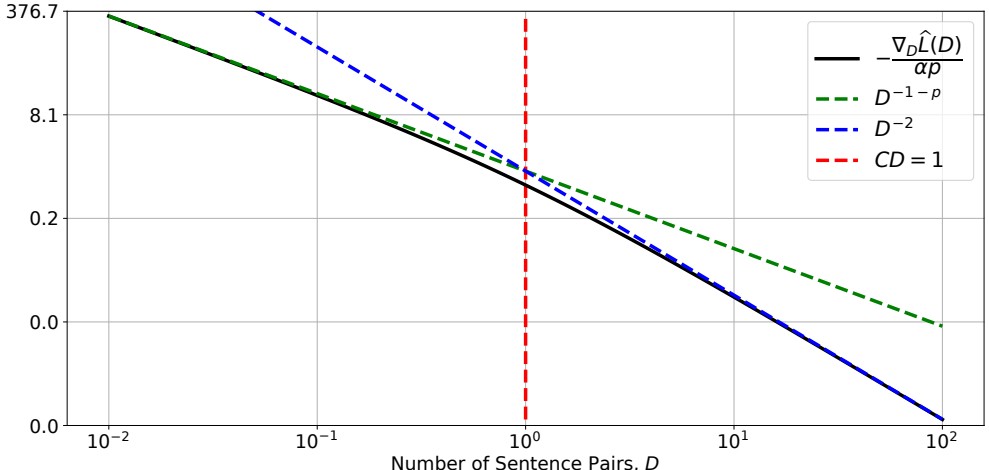

Figure 7: Phase transition from data-limited regime to model-limited regime. For simplicity, we assume $D_0 = 1$.

## D  SCALING LAWS WITH DIFFERENT ARCHITECTURES

The fitting parameters for Figure 1B are shown here.

|  | $\alpha$ | C | p |
|---|---|---|---|
| Encoder-Decoder | 1.969 | 0.057 | |
| Decoder-only | 1.817 | 0.11 | 0.285 |
| Hybrid-LSTM | 2.011 | 0.078 | |

We now show additional plots for decoder-only and transformer-LSTM hybrid models, with individual fits for each model.

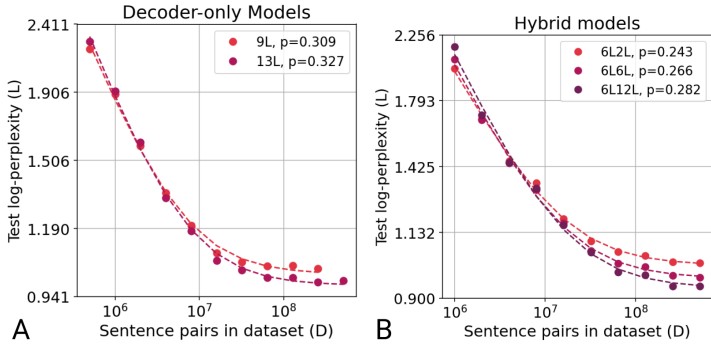

Figure 8: Separate fits for architecture with different depths (A) Decoder only (B) Transfomer-LSTM Hybrids

