# OpenReview forum: "Data Scaling Laws in NMT: The Effect of Noise and Architecture"
_ICLR.cc/2022/Conference — ICLR 2022 Submitted_

### Official Review · Reviewer_Pwfc · 2021-11-01

**Correctness:** 2
**Technical Novelty And Significance:** 2
**Empirical Novelty And Significance:** 2
**Recommendation:** 3
**Confidence:** 4

**Main Review:**

Reasons for score:

I have major concerns about the contributions this paper adds compared to [1] and [2]. Most of the conclusions in the paper have already been confirmed by these previous works.

- The scaling law of dataset size has also been thoroughly studied and verified by [2]. The form of scaling law used in this paper is the same as what in [2] except an extra $L_\infty$ term to control the convergence.
- [2] has already shown the effect of larger models on data scaling laws with models with different numbers of parameters. [1] has also thoroughly studied the effect of different model sizes on the scaling laws of neural machine translation models.

In addition, some conclusions from the paper are also not convincing:

- The paper only conducts experiments on a single language pair (English - German), which doesn't necessarily mean all the patterns found in the paper would generalize to other datasets.
- When testing for different models, there is only one data point for a type of model at a specific dataset size. However, the performance of a model is determined by lots of other hyper-parameters like the choice of the optimizer, the learning rate, dropout rate, etc. The authors didn't discuss how they decide these hyper-parameters and how they affect the scaling law.
- In Figure 3A, the curve of the no-filter case misses the data points a lot. Also, while the three cases in this figure obviously have different test log-perplexity for the largest dataset case, the fitted curve converges to the same point, and in the main text the authors claim that "Loss at convergence $\alpha C^p$ is the same". This conclusion is based on errored fitted curves and is consequently wrong.

Other comments:

- "However, recent trend in machine learning largely confirms the trade-off between architectural biases and the amounts of data, namely tendency towards architectures with low inductive biases are being trained on large amounts of data, compute and parameters." Please add references for this claim.

Minor Comments:

- On page 4, please define $\beta, N_e, N_d$ instead of only referring to previous works.

**Summary Of The Paper:**

This paper verifies the scaling laws for neural machine translation models in terms of dataset size. In addition, with different model sizes, model architectures, and different noise level of the training data, the scaling law still holds. This means the effect of suboptimal model architectures and noisy data can be compensated with simply adding more data.

**Summary Of The Review:**

The contributions of this paper are incremental and the conclusions are also not convincing.

---

> ### Author Response · Authors · 2021-11-20
> **Response to Reviewer Pwfc**
>
> We thank the reviewer for their comments. We have addressed their concerns below:
>
> Comparison to prior work: We kindly refer the reviewer to the common response to all reviewers for a detailed comparison of our work to prior work, and what our additional contributions are. Most importantly, we study the effect of changing various aspects of the training setup such as architecture, noise, filtering and back-translation, none of which have been studied in either of the stated prior works.
>
> Single language pair: We kindly refer the reviewer to the common response to all reviewers. In summary, while studying the behavior of multiple language pairs would be an interesting direction for future work, the focus of our work is to exhaustively study how different training choices affect the scaling curves.
>
> Hyperparameters: We apologize for missing details about hyperparameter optimization from the paper. We will include these in the next revision. We optimized the hyperparameters for the models in the large data regime (~ 2B data points) on the original dataset, and used the same hyper-parameters for all the dataset sizes. So, we consider the hyperparameters (dropout rate, learning rate, logit clipping etc) as a part of the model architecture choice. Thus, our results show that for a combined choice (model + hyperparameter) we have a scaling law with similar exponent across many settings. We would like to emphasize that we did not cherry-pick hyperparameters to match the slopes --- the hyperparameters for all models (encoder-decoder, hybrid, decoder-only) were fixed on the large data regime for the clean dataset before we conducted any further experiments for varying dataset sizes. Thus, our results showing that the exponent is similar for this combined choice (model + hyperparameter) is still significant.
> For a more complete picture, one would produce the scaling laws for different hyperparameter choices and take the pareto-optimal curve over these hyperparameter choices. However, producing this entire grid is computationally prohibitive.
>
> Figure 3A: We respectfully disagree that the fitted curve missed the experimental results by a large amount. Some deviations from the fitted curve are to be expected, as has also been seen in many prior works on scaling laws (Kaplan et. al https://arxiv.org/abs/2001.08361, Henighan et. al. https://arxiv.org/abs/2010.14701). Moreover, the sub-optimal fit occurs in the intermediate region between the ‘data-limited’ and ‘parameter-limited’ regions, whereas the exponent is dictated primarily by the fit in the data-limited region <=16M data points (as we discuss in more detail in Appendix C) where we find a good fit with the experimental results. Re convergence to the same loss value, the difference between the two points is 0.02 experimentally, which is close to the range of the standard deviation over different random seeds as shown in Figure 4B. We would like to emphasize that the log-log plot amplifies the differences between these points.
>
> References for inductive bias-data claim: We will include references in the next update. Broadly, we mean that similar architectures (such as transformers) have been successful in NLP, vision and other domains showing that the architectures with low bias adapt well across different domains.

---

### Official Review · Reviewer_Nuk2 · 2021-11-02

**Correctness:** 3
**Technical Novelty And Significance:** 3
**Empirical Novelty And Significance:** 3
**Recommendation:** 6
**Confidence:** 4

**Main Review:**

**Strengths:**

- The paper has strong empirical experiments at scale to validate the scaling laws. Compare with previous work (Gordon et. al), the training data is much larger, up to 512M sentence pairs for English->German. While I believe that the models used in the paper have a significantly large number of parameters, I couldn’t find this information in the paper. Could the authors elaborate the size of these models?

- There are some interesting findings in the paper when scaling up the training data. For example, it seems that data filtering can be replaced with more unfiltered data.

- The experiments with data noise are carefully designed and controlled. The results provide useful insights.

**Weaknesses and Questions**

- As the authors already called out in the paper, one major weakness I found is the use of log-perplexity as a proxy for language generation quality. *To what extent would improvement in test log-perplexity reflect in human evaluation (HE) and would the improvement in human evaluation follow dismissing returns laws as the amount of data increases?* Although the authors discussed about another potential metrics (e.g., COMET, BLEURT) I wonder if the authors have consider using human evaluation for one of those model (e.g., 6L6L) at different data scales? If yes, what prevented it from being done?

- While the paper focuses mostly on test log-perplexity in scaling laws, I think there are other aspects of translation quality that are important for making deployment decisions. Larger models trained on large amounts of data could potentially exacerbate some known issues of NMT such as gender-biases. I understand that this is not the main focus of the paper, but I think it might be useful to have this angle in the discussions when deployment decisions are rarely made based on perplexity and BLEU scores.

- As the training data is crawled from the web, I imagine that some part of the data could have translationese source. I wonder if the tetsets used in the work are source-original? Perhaps, my question would be *what the graph of fitting the scaling laws look like when evaluating on translating in the natural direction?*

- While all the models in the paper are dense model, I wonder if the scalling laws hold for sparse models (e.g, Sparse Mixture of Experts)

- What is the value of $L_\infty$? Do you set it fixed for fit it?

**Few recommendation for paper presentation:**
- In the paragraph below equation 2, I think you meant 6L2L as a held out model, not 6L28L
- In subsection 4.2.1 at the end of the first paragraph, I think you meant table 2 (i.e.The results are shown in Figure 1C and Table 3.)
- Table 3 isn’t mentioned in subsection 4.2.2 DEPENDENT NOISE: BACK-TRANSLATION
- If possible, I would recommend putting the figures where they are discussed.
- In equation 2, are $p_e$ and $p_d$ are free parameters? They are not mentioned in the following paragraph.
- I think it's worth to explain $L_\infty$.
- Some part of the paper should have been proof-read and cleaned up. In the implication paragraph, could the author ellaborate where the addtional amount of data $O(D^{−5/4}) $ and $O(D^{-2})$ come from?


**Summary Of The Paper:**

The paper provides an empirical study of scaling laws for neural machine translation with respect to data noise (filtered/unfiltered/back-translation) and model architectures (transformer encoder-decoder, transformer decoder, transformer-encoder LSTM decoder). Focus on data aspect in scaling laws, the paper carried out a series of experiments on multiple data scales ranging from low-resource (500K sentence pairs) to large scale in industrial setup (512M sentence pairs)

Scaling Laws: the paper shows that their scaling laws (in equation 1) can be fit five different architectures (2L6L, 6L2L, 6L6L, 28L6L, 6L28L) on in-domain testsets and 11 out-of-domain testsets using 6L6L models. The scaling laws also can be used to predict the test log-likelihood of an unseen model (Figure 2A).

Architecture: Experimenting with two different encoder-decoder architectures, namely transformer encoder-decoder and transformer-encoder LSTM decoder, the paper shows that marginally worse architecture can be boosted up by adding more data.

Data noise: Throughout a series of control experiments, the experiments show that data filtering can be replaced by adding more unfiltered data. The scaling laws can fit both independent noise and dependent noise (via back-translation)


**Summary Of The Review:**

The paper shows strong empirical results on fitting scaling laws. The experiments are carried out on large scale dataset and large models. While this provides valuable insight for practictioners on making decision when training a NMT system, using log-perpelxity without human evaluation is problematic.

---

> ### Author Response · Authors · 2021-11-17
> **Response to Reviewer Nuk2**
>
> We thank the reviewer for their comments. We have tried to address the remaining concerns below.
>
> “Number of parameters”: Our experiments consist of model sizes ranging from ~100M (6L2L) to ~900M (6L28L). We will update the revision with these details.
>
> “Human evaluations”: We thank the reviewer for their suggestion. We agree that understanding the relationship of log-perplexity with human evaluations is an important question. However, our budget mostly limited us in conducting detailed human evaluation experiments. Moreover, the methodology for performing human evaluations itself remains an active and contested research area https://arxiv.org/abs/2104.14478. The feasibility and affordability of resorting to a reliable human evaluation metric (e.g. MQM) for our work, which is answering multiple research questions related to data, was unfortunately beyond our reach and capacity.  Thus, we believe that studying the relationship between log-perplexity and human evaluations is a research problem that is not specific to data scaling, and hence out of scope of our work.
>
>
> “Source original test-sets”: The test set used for a majority of our figures is a held-out test set from the training distribution, which consists of a mix of source-original (majority) and target-original data. However, we evaluate our models on a variety of different test sets with different compositions. As Figure 6 shows, we found that the test set composition does not have a significant impact on the data scaling exponents. Moreover, the performance of different test sets seems to have an almost-linear relationship (Figure 6C).
>
> “Sparse Mixture of Experts”: Thank you for the suggestion, this would be an interesting direction for future work!
>
> “L_infinty”: L_infinity is a fixed parameter that was found by Ghorbani et. al. for parameter scaling. We borrow their parameter scaling law as is, and plug it into the right side of Equation 2. In their work, they fit this L_infty by training a series of models on 2B data points. We apologize for the lack of clarity, and we will update the paper to explain this more clearly.
>
> We are also grateful for the recommendations on paper presentation, we will edit the paper to accommodate these suggestions.

---

### Official Review · Reviewer_fUfX · 2021-11-06

**Correctness:** 3
**Technical Novelty And Significance:** 2
**Empirical Novelty And Significance:** 2
**Recommendation:** 5
**Confidence:** 3

**Main Review:**

Strengths:
- With the recent advances in large language models, scaling laws for NMT is timely and important.
- A large number of empirical studies are performed to investigate the impacts of different model & data choices on the final NMT scaling laws.
- The depicted scaling laws could provide empirical guidance to the NMT community.

Weaknesses:
- The studies are mostly focused on one language pair (English-to-German), there isn't clear evidence whether their findings could generalize to different language families.
- Since the paper is mostly an empirical work, there's lack of novelty in methodology contributions. However, I am not sure if we should count too much on the methodology contribution for such scaling laws papers.
- Some findings don't seem very informative, e.g., the scaling laws on model sizes and model family.
- The following paper seems very relevant, which should be cited:
Scale Efficiently: Insights from Pre-training and Fine-tuning Transformers

**Summary Of The Paper:**

The paper studies data scaling laws for NMT. Specifically, it investigates the following factors empirically:
- model architecture
- model sizes (different numbers of encoder or decoder layers)
- data filtering
- different types of manually-added noises.

The paper has performed a large number of studies to demonstrate some empirically-useful findings for training the NMT models.

**Summary Of The Review:**

The major takeaways for the paper, I think, are:
- Changing in model sizes and architectures have very similar scaling patterns, with a shared component.
- Different noise reduction filtering methods (e.g., CDS and Bicleaner) can also be fit into the shared component.
- Adding different types of noises can dramatically downgrade the NMT performance, while independent noises can still fit into the share the component, except the dependent noises via back-translation.

Overall, these findings emphasizes again on the importance of data quality for NMT models, which is somehow known to the community. The paper further quantified such importance with the data scaling laws.

---

> ### Author Response · Authors · 2021-11-17
> **Response to Reviewer fUfX**
>
> We thank the reviewer for their comments and for recognizing that our study has the potential to provide empirical guidance to the community. We have addressed the raised concerns below:
>
> “Multiple language pairs”: We kindly refer the reviewer to the common response posted above for a detailed discussion on our reasoning to focus on a single language pair.
>
> “Methodological contributions”: While the methodology of changing dataset sizes and fitting a scaling law is not novel, we would like to emphasize that the experimental design choices are. We are the first to study how changes in training _change_ the scaling law, as opposed to just establishing that a scaling law exists. Moreover, we show how practical insights can be gained by using a scaling approach.
>
> “Some findings don’t seem informative”: Could the reviewer please clarify which experiment they mean by this? We have summarized the takeaways below, but we would be happy to clarify further:
> Model size: Our main contribution here is to show that the asymmetric models (encoder-heavy vs. decoder-heavy) have similar data scaling exponent.
> Model architecture: We show that despite architecture (transformer vs. LSTM) and task setup (LM vs. supervised in decoder-only transformers) differences, which could imply different ‘inductive bias’, the data scaling law shares an exponent, indicating a deeper similarity between these different architectures.
>
> “Scale efficiently”: The paper in question studies scaling with respect to parameters and computation, while we study the scaling with respect to data. Therefore, we did not include it in our initial list of related works. We will be happy to add an extended related works section in the appendix that reviews parameter/compute scaling work in addition to data scaling work.
>
> “Data quality importance is known”: We agree that the community is well aware that the quality of data plays a role in the performance of the model. We would like to further emphasize that the manner in which quality affects performance may be different from prior intuitions - for instance, we find that filtering _does not_ change the exponent of the scaling curve, implying that noise is not catastrophic and can be compensated for by adding more data.
>
> As the reviewer also mentions, the main takeaways from our work are the impact of changing architecture and noise in the training setup. We believe that our findings in this regard are novel and useful, both practically for making decisions for large-scale model training, as well as theoretically for understanding why the exponent remains unchanged in so many situations.

---

### Official Review · Reviewer_XWAu · 2021-11-07

**Correctness:** 4
**Technical Novelty And Significance:** 3
**Empirical Novelty And Significance:** 3
**Recommendation:** 6
**Confidence:** 4

**Main Review:**

Pros:

1. Well executed empirical study that complements the work of Ghorbani et al. 2021 by considering different architectures such as transformer LMs, transformer-LSTM hybrids, and data filtering along with results on publicly available web-crawled corpora.

2. The paper offers clear suggestions on when to use noisy data based on available compute and model sizes.

Cons:

1. A lot of the analysis and scaling results are already present in Ghorbani et al. 2021 which this paper borrows from. For example, 1) the asymmetric encoder-decoder scaling 2) OOD test set results on almost identical test sets 3) backtranslated vs forward translated data in this paper vs “original src” and “original tgt” in Ghorbani et al etc.

Questions:

1. The paracrawl.eu website reports a lot less than 750M En-De sentences for Paracrawlv8. It would be best to report the exact source from which this data was downloaded.


**Summary Of The Paper:**

The paper studies model and data scaling laws for Neural Machine Translation. It studies how the test cross-entropy loss scales with 1) Different seq2seq model architectures (transformer seq2seq, transformer LM and a transformer-LSTM hybrid) and asymmetric scaling of the encoder and decoder 2) (a) Training data size and composition - backtranslated data from models of varying size, filtered data using cleaning tools, synthetically introduced noise in the source and target languages (b) In-domain vs out-of-domain test data.

**Summary Of The Review:**

Overall this is a very thorough empirical study of NMT scaling that is sure to benefit the NMT research community immensely when read alongside Ghorbani et al. 2021.

---

> ### Author Response · Authors · 2021-11-17
> **Response to Reviewer XWAu**
>
> We thank the reviewer for their comments and for recognizing that the work can benefit the NMT community. We have provided a detailed comparison of Ghorbani et. al. in our common response to our reviewers. In summary, while the setup in both papers shares similarity, the qualitative insights from both works differ.
>
> Paracrawl data: We apologize for the error. We use the raw German dataset (1.4TB with 26B sentence pairs) provided at this link https://paracrawl.eu/v8 We will add an appendix section in the next revision that details the data processing steps.

---

> > ### Comment · Reviewer_XWAu · 2021-11-25
> > **Paracrawl provides bicleaner filtered data**
> >
> > Thanks, I'm still not sure where one can obtain the raw dataset containing 26B sentences. The link above only seems to contain only the bicleaner filtered data of 261M sentences.

---

> > > ### Author Response · Authors · 2021-11-30
> > > **Using Raw Data from dropdown menu**
> > >
> > > The 26B sentence pairs refers to the 1.4TB 'Raw' data at the following link https://s3.amazonaws.com/web-language-models/paracrawl/release8/en-de.classified.gz . To obtain this, follow https://paracrawl.eu/v8 -> "German" -> click on the arrow mark next to 'TXT'. This gives a dropdown menu with for the raw data. Please see the screenshot https://pasteboard.co/G7KrNQGijyrX.png We hope this clarifies your question!

---

### Author Response · Authors · 2021-11-17
**Response to all reviewers addressing common concerns**

We thank all the reviewers for their comments. Below we address two concerns that were raised by multiple reviewers.


1. **_Similarity to prior work_**: Several reviewers have pointed out certain points of similarity between Gordon et. al. [1] and Ghorbani et. al. [2] We detail our contributions in relation to these works below. We will update the next revision with a summary of this discussion.

     **Ghorbani et. al.**: While the experimental setup in both works share similarities, we believe the two papers are different in scope and offer qualitatively different insights.  The main difference between [2] and our current work is that we study scaling with respect to _dataset size_, while [2] focuses on scaling with respect to the number of parameters. Since the number of parameters and dataset size have no trivial correspondence (except in simple cases such as linear models), we cannot assume that insights from parameter scaling will transfer to data scaling and vice-versa. For example, in [2] the authors found that composition of the test set (source original vs. target original) strongly affects the scaling exponents in parameter scaling. In contrast, data scaling does not exhibit such a difference and performance on all test sets scales with a similar exponent (See Figure 6 in Appendix B). Moreover, since parameter scaling depends on the specifics of the architecture and the parameterization of the network, different architectures are not expected to have similar parameter scaling (as also observed by https://openreview.net/forum?id=Wrtp36cbl61), but as our work shows, they have similar data scaling.

    **Gordon et. al.**: This work studies both data and parameter scaling for Neural Machine Translation, and appeared concurrently during our investigation into NMT data scaling laws. We believe that this work is complementary to ours, in that it confirms the usefulness of scaling laws in understanding NMT systems. However, we believe that the papers focus on very different aspects of the problem, which we discuss below:
(i) _Studying the effects of changing training setup_: While [1] establishes the existence of a data and parameter scaling law, the main focus of our work is to understand which aspects of the training setup _change_ the data scaling curves. To this end, we conduct detailed experiments that change the architecture, filtering, synthetic noise and back-translation, while these aspects are kept constant in [1].
(ii) _Scale of experiments and resulting difference in scaling law_: The two papers conduct experiments at data and parameter scales that differ by more than an order of magnitude. The experiments conducted in [1] go up to a maximum of 50M sentence pairs, while our experiments range up to 1B data points, and the maximum model sizes are up to ~50M parameters vs. 900M parameters respectively. Consequently, the scaling laws observed in [1] and our work have subtle differences in their form: Our scaling law consists of an additional L∞ term  that remains even when the Ne , Nd , D = ∞. We believe this difference arises because Gordon et al. (2021) use relatively smaller model sizes . We provide a more detailed discussion of this difference in Footnote 2 in our work. Working in the smaller data regime allows [1] to make better predictions about the behavior of NMT systems for low-resource languages, while working in the large data regime in our work allows us to assess the effectiveness of interventions such as filtering (where having a large dataset allows us to throw away data with filtering and still observe reasonable scaling laws). Thus, we believe that these two papers complement each other in providing a more holistic picture of scaling in NMT.

---

> ### Author Response · Authors · 2021-11-17
> **Common response continued**
>
> 2. **_Single language pair_**: While we agree that conducting experiments with other language pairs can be additionally informative, we believe that our current large scale study already has the potential to yield important insights. In a computationally intensive basic research study like ours, we have to make hard experimental trade-offs to stay within our computational budget. We would like to emphasize that the experiments conducted in our work range in model sizes from 100M-900M parameters, and from 500K-500M dataset sizes, involving a significant amount of computational resources. Thus, for this work, we chose to focus on exhaustively studying a single language pair to understand how different training choices (architecture, noise, filtering) impact the data scaling curves. We chose to study the En-De language pair since it is most widely used and is believed to be indicative of the behavior of high-resource language pairs.
>
>    Additionally, concurrent work Gordon et. al. has shown that data scaling laws do arise for other language pairs and have similar functional form. Since our choice of training interventions, e.g. the choice of architectures, the type synthetic noise or back-translation modelling, do not have any specific relationship to the choice of language pair, the choice of language pair may not qualitatively impact the results we find. That is, we expect the results to have a similar qualitative similarity like having similar exponents with architecture changes, but could change specifics like the values of the scaling exponent and intercepts at least in the high resource Indo-European language setting.
>
>     While studying how the nuances of different language pairs (such as different vocabulary or language difficulty) impact the scaling curves is indeed interesting and can be impactful, we believe that our current results on one language pair already show certain surprising behaviors that merit further research. Thus, given the scale of our study we chose to summarize our results at this stage to share it with the community. We hope that future research will build on this setup to provide a clearer picture!

---

### Decision · Program_Chairs · 2022-01-20

**Decision:**

Reject

**Comment:**

This paper analyzes the data scaling laws in NMT tasks with different network architectures and data qualities. The main purpose of this paper is to investigate how such different experimental setup affects the scaling law. The authors found that those difference does not have strong impact on the scaling exponent, and a small difference of model architecture and data noise can be compensated by larger data size.

This paper gives nice justification of data scaling law from some different aspects which is instructive to some extent. On the other hand, the paper has some weakness as listed in the following: (1) The scaling law itself has been analyzed by many papers, and its novelty is rather limited. I acknowledge that this paper investigates different aspects of the data scaling law and the size of experiments are larger than existing work. However, the result is rather unsurprising. (2) The experiments are conducted mostly on one language pair (English-to-German), it is still unclear whether the findings are universal to other language pairs. As the authors responded, exhaustive experiments over all language pairs are unrealistic but some more investigation to more general data sets could be conducted to strengthen the paper.

This paper is around the borderline. Some reviewers were rather positive to this paper. However, they also pointed out the concerns I listed above and they do not show strong support on the paper.
In summary, although this paper shows some instructive findings, it is still a bit below the threshold of acceptance.